# Establishment and Application of a New Parameter Model for Quantitative Characterization of the Heterogeneity of Thick, Coarse-Grained Clastic Reservoirs: A Case Study of the Badaowan Formation in the Western Slope of the Mahu Depression, Junggar Basin, China

**Boyu Zhou [1], Xiaodong Zhao [2,*], Xuebing Ji [2], Xinyu Wu [1], Wenping Zhao [1] and Xi Rong [2]**

[1] Exploration and Development Research Institute, Xinjiang Oilfield Company, China National Petroleum Corporation, Karamay 834000, China; fczby@petrochina.com.cn (B.Z.); wuxinyu@petrochina.com.cn (X.W.); zwp@petrochina.com.cn (W.Z.)

[2] School of Petroleum, China University of Petroleum (Beijing) at Karamay, Karamay 834000, China; 15832835908@139.com (X.J.); 18032951936@163.com (X.R.)

* Correspondence: zhaoxd2019@cupk.edu.cn

**Abstract:** The rock composition of thick-layer, coarse-grained clastic reservoirs is complex. There are large variations in granularity and poor selectivity. Reservoirs of thick-layer, coarse-grained clastic rocks are extremely heterogeneous. Current conventional parameters for quantitative characterization of reservoir heterogeneity, such as the calculation values of the permeability variation coefficient, the permeability rush coefficient, and the permeability contrast, are unbounded, have different representation angles, and the quantification degree of the characterization method is not high. This study takes the thick layer of the coarse-clastic rock reservoir developed in the western slope of the Badaowan Formation in the Mahu Depression of the Junggar Basin as an example. Through core observation, microscopic characteristics, and analysis of laboratory data, a new quantitative characterization parameter of heterogeneity is proposed, and a reservoir interpretation parameter model is established. The results were as follows. (1) The pore development of the thick, coarse-grained clastic rock reservoir is complicated, the sorting and pore structure are poor, the reservoir heterogeneity is strong, and the permeability has double peaks. (2) We propose a new parameter to evaluate reservoir heterogeneity: the fluctuation a coefficient. This essentially compares the average permeability of two adjacent layer sites with the average permeability. The fluctuation coefficient can reflect the fluctuations in permeability, and the larger the fluctuation coefficient, the stronger the heterogeneity. In addition, it has the advantages of a clear characterization target, bounded calculation data, and the same characterization angle, etc., thereby realizing the quantitative characterization of the macro degree of reservoir heterogeneity under a unified standard. (3) This parameter was used to evaluate the reservoir heterogeneity of the Badaowan Formation in the western slope of the Mahu Depression. Most wells in the study area had a fluctuation coefficient of about 0.3, but others ranged between 0.2 and 0.6. It is concluded that the larger the fluctuation coefficient of the study area, the better the oil content because these types of reservoirs have strong heterogeneity. The fluctuation coefficient can effectively reflect the strength of the heterogeneity and can also provide a reference for further reservoir enrichment research.

**Keywords:** Junggar Basin; Mahu Depression; thick coarse-grained clastic rock reservoir; heterogeneity; fluctuation coefficient

## 1. Introduction

Thick-layer, coarse-grained clastic rock reservoirs generally vary in size from several meters to more than ten meters. Their rock composition is complex and diverse, with large

variations in rock particle size and low maturity. In addition, sorting is poor, longitudinal physical property differences are large, and the sedimentary structure is complex. These features make thick-layer, coarse-grained clastic rock reservoirs highly heterogeneous [1,2]. This type of reservoir differs from the conventional clastic rock reservoir, and the results from conventional evaluation methods are poor, which has significant impact for oil and gas exploration and development [3]. In previous research concerning reservoir heterogeneity, only a single parameter has been used, such as the permeability variation coefficient, the permeability rush coefficient, or the permeability contrast characterization methods. As they are based on actual oilfield development situations, these parameters describe reservoir heterogeneity from different angles and, because of this, have significant limitations: blind spots in characterization and an inconsistent degree of heterogeneity leading to a weak quantitative characterization ability, which has been unable to meet all the requirements [4,5]. Thick, coarse-grained clastic rock reservoirs are a very important reservoir type, but because of their strong heterogeneity and special pore structure, it is difficult to establish a calculation model of the related parameters [6]. However, this type of reservoir has great development potential, and its evaluation and characterization is crucial.

Great importance has been attached to the study of reservoir heterogeneity at home and abroad, and there is a commitment to quantitative characterization research in this field [7]. To characterize reservoir heterogeneity, scholars use geostatistical theory to calculate and compare characteristic parameters of regional variations in permeability planes, permeability distribution characteristics, and various heterogeneity parameters. Some scholars also use two-dimensional fluid flow models, two-dimensional permeability models, and petrophysical models to characterize reservoir heterogeneity semiquantitatively [8–11]. Other scholars use multiple parameters to characterize the degree of heterogeneity from the interlayer, intralayer, plane, and microscopic perspectives [9–12]. Common evaluation parameters include the permeability variation coefficient, permeability rush coefficient, and permeability contrast. As they are based on actual oilfield development situations, these parameters describe the heterogeneity of reservoirs from different perspectives and have significant limitations [13–15]. Shortcomings include blind spots and an inconsistent degree of heterogeneity, which mean that the ability to quantitatively characterize heterogeneity is relatively weak. There is, therefore, an urgent need to propose new parameters for heterogeneity characterization, which are more comprehensive and quantitative. In recent years, the development and combination of various disciplines, including nongeological disciplines, such as fuzzy mathematics, economics, and statistics [16,17], has resulted in the use of heterogeneous composite indexes to characterize reservoir heterogeneity: the Theil index, logging data, heterogeneous composite index, reservoir modeling, Lorentz curve, and gray set pair analysis evaluation methods [18–20]. Of these, the entropy weight method ignores the importance of the index parameters themselves, which is a common problem with objective weighting methods. It is only applicable to the study of reservoir heterogeneity when there is little difference in the values of several parameters but significant difference in the values of others. The defect of the gray correlation analysis method is that if there are differences in the reference sequence, normalization method, and resolution coefficient; the correlation will not be unique. The problem of information duplication caused by the correlation of indicator information is difficult to resolve, and therefore, the selection of indicators has a significant impact on the evaluation results. Although AHP fully considers and integrates a variety of qualitative and quantitative information in a comprehensive evaluation, in its practical application there remain issues of randomness, subjective uncertainty of evaluators, and fuzziness of evaluation conclusions. Such subjective assumptions inevitably reduce the credibility of results, and the discriminant matrix is prone to serious inconsistency. Few of these methods are suitable for the characterization of thick, coarse-grained clastic rock reservoirs with strong heterogeneity. In this study, characterization parameters for the heterogeneity of thick, coarse-grained clastic rock reservoirs were fully investigated in order to obtain a suitable characterization method.

## 2. Limitations of Conventional Heterogeneous Characterization Parameters

For a long time, the main parameters used to quantitatively characterize the degree of heterogeneity of sand bodies have been the permeability variation coefficient, permeability rush coefficient, permeability contrast, etc. [21,22]. The permeability variation coefficient is used to measure the degree of variation of the longitudinal permeability value of the sand body relative to its average value. At present, it is the most widely used heterogeneity characterization parameter. However, in the specific application process, the value of this parameter is theoretically distributed from 0 (homogeneous) to ∞ (extreme heterogeneity), is unbounded, and cannot accurately express the heterogeneity of the reservoir. The permeability rush coefficient represents the ratio of the maximum permeability of the sand to its average permeability, and its shortcoming mainly lies in its inability to characterize the thickness and scale of the maximum permeability interval [23]. The permeability contrast parameter represents the ratio of the maximum to the minimum permeability in the sand layer. This parameter does not consider the thickness and average permeability of the sand body, so the size of the permeability contrast cannot reflect the degree of reservoir heterogeneity. Even if they have the same permeability contrast, the actual degree of heterogeneity may be completely different in reservoirs of different thicknesses. Clearly, there are defects in these quantitative characterization parameters of reservoir heterogeneity which limit their application scope [24]. At present, the permeability variation coefficient, permeability rush coefficient, grade difference, and other parameters used to quantitatively characterize the degree of reservoir heterogeneity are all based on actual oilfield development, and consequently, reservoir heterogeneity is characterized from different angles and with different focuses. However, from the perspective of the algorithms of the aforementioned parameters, because the distribution of the calculated values is theoretically unbounded, they are of limited use for quantitative evaluation of the degree of heterogeneity [25,26]. The compromise approach is to divide the calculated values into several classes so that the degree of heterogeneity is divided into different levels, but because of the strong subjectivity of classification schemes, these produce different evaluation results for the degree of heterogeneity [27].

Each parameter also has defects in specific applications. The permeability variation coefficient is used to measure the degree of variation of longitudinal permeability in sand relative to its average value, and at present, this is the most widely used heterogeneous characterization parameter. However, for two sand bodies with similar permeability variation amplitudes and large differences in mean permeability, the permeability variation coefficient calculated for the sand body with low mean permeability is always larger. In other words, the permeability variation coefficient is greatly affected by the mean permeability in the algorithm, and the characterization of the degree of heterogeneity is relatively ignored (Figure 1a). The permeability rush coefficient represents the ratio of the maximum to the average permeability of sand. Because the permeability variation process is ignored in the parameter calculation, the breakout coefficient calculation may give the same result for reservoirs with different degrees of heterogeneity (Figure 1b). The permeability contrast parameter represents the ratio of maximum to minimum permeability in sand. This parameter emphasizes the absolute difference between the maximum and minimum permeability values but ignores the change process between the maximum and minimum permeability values. As a result, the magnitude of the step difference is unable to reflect the degree of reservoir heterogeneity. For reservoirs of different thicknesses, even if the calculated step difference is the same, the impact on reservoir development may be completely different (Figure 1c). In summary, the above heterogeneity characterization parameters have different features and application emphases in the characterization of reservoir heterogeneity, but they are relatively weak in the quantitative characterization of the macro degree of reservoir heterogeneity [28,29].

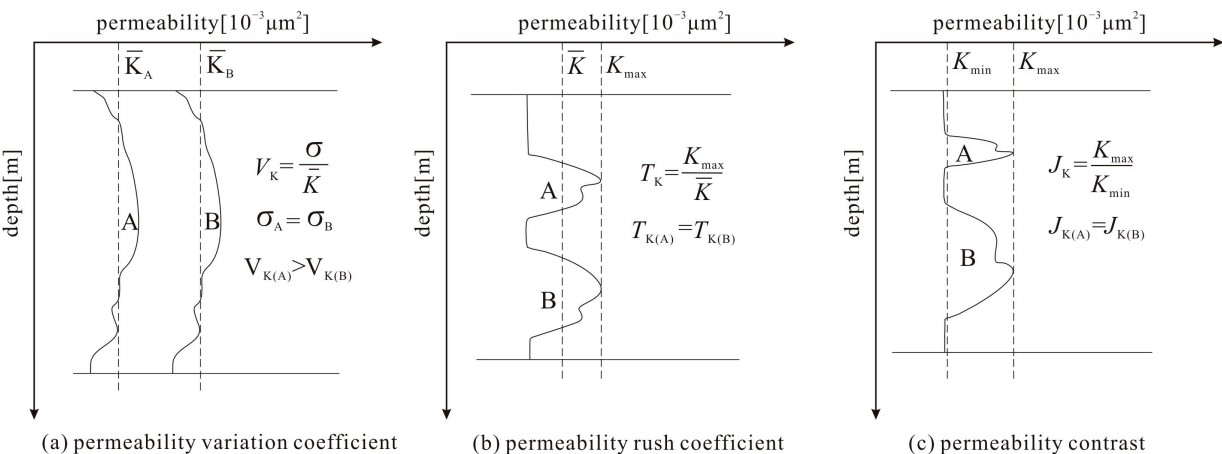

**Figure 1.** Defects in quantitative characterization parameters of reservoir heterogeneity.

Different combinations of permeability at each point of the reservoir reveal the different heterogeneity characteristics of the reservoir. However, when two sets of data with the same thickness and the same single point permeability are in different combinations, the permeability variation coefficient, permeability rush coefficient, and permeability contrast calculated according to their formulas are the same. Therefore, a new parameter is required to overcome problems, such as large variations and irregularities in single well permeability.

## 3. Materials and Methods

### 3.1. Materials

In order to study the characteristics of thick, coarse-grained clastic rock reservoirs and evaluate the heterogeneity of the reservoir, this paper describes the macro characteristics of the reservoir primarily through core observation, qualitatively and semi-quantitatively describing the pore throat structure of the reservoir using thin-cast scanning electron microscopy (SEM) and quantitatively characterizing the pore throat structure via a mercury injection experiment. In this study, 276 samples were selected from the Aihu 501, Ma 625, Ma 606, and Bai65 wells (note: Aihu, Ma, and Bai are the names of the wells), and the samples were located in the first member of the Jurassic Badaowan Formation. The rock slice, cast slice, scanning electron microscope identification, and determination of porosity and permeability were carried out for each of the samples. In order to quantitatively evaluate the pore structure of the reservoir, 12 samples from Aihu 501 were selected for mercury injection determination. The cast thin sections, SEM images, and analytical data were completed in the Geological Laboratory of the Xinjiang Oilfield Company, PetroChina. The mercury injection experiment was completed at the Karamay Campus, China University of Petroleum (Beijing). The mercury injection experiment was completed using an HD-505 (mercury intrusion meter model) a high-pressure porosity structure instrument. The ambient temperature was 18 °C, the relative humidity was 35–50%, the maximum mercury inlet pressure was 30 Mpa, and the corresponding pore throat radius was 0.025 μm.

### 3.2. Methods

#### 3.2.1. Parameter Establishment Process

Because of its different characteristics compared with conventional reservoirs, the permeability of thick, coarse-grained clastic rock reservoirs may vary greatly at each layer site. If the average permeability of two adjacent layer sites is compared with the overall average permeability, then the permeability fluctuation between layers can be obtained. Compared with conventional clastic reservoirs, the most prominent feature of thick, coarse-grained clastic reservoirs is that the reservoir thickness is large and stable, and the number of layers should also be used as a parameter standard in the evaluation. The

microheterogeneity is closely related to the development effect of the reservoir and is also the root cause of the macroscopic seepage phenomenon. Permeability is a crucial parameter which can effectively reflect changes in the reservoir. Therefore, permeability is selected as the evaluation parameter, and a new characterization parameter, wave coefficient, which reflects the characteristics of thick, coarse-grained clastic rock reservoirs, is proposed to evaluate reservoir heterogeneity.

3.2.2. Parameter Proposal

In the case of the same thickness and the same permeability at a single point, the fluctuation coefficient is the ratio between the average difference of two adjacent permeability data points in the formation and the average permeability, because different combinations of permeability at each point indicate the different heterogeneous characteristics of the reservoir.

$$v_b = \frac{\sum_1^n |k_i - k_{i+1}|}{(n-1)\bar{k}} \tag{1}$$

$v_b$ represents the fluctuation coefficient (dimensionless), $\bar{k}$ represents the average permeability ($10^{-3}$ um$^2$), and $k_i$ represents the value of the $i$ th permeability data point ($10^{-3}$ um$^2$). Both $i$ and $n$ are positive integers.

The smaller the permeability fluctuation coefficient, the smaller the difference in permeability between adjacent reservoirs, and the weaker the reservoir heterogeneity; and vice versa. The fluctuation coefficient adds the irregularity of permeability into the evaluation of reservoir heterogeneity to improve the accuracy of the reservoir heterogeneity evaluation. Thick, coarse-grained clastic rock reservoirs differ from conventional reservoirs in that permeability can vary significantly at each layer location. By calculating the change degree of adjacent points using Formula (1), the variation of permeability between points can be reflected and the heterogeneity of reservoir evaluated. This parameter can be used together with other conventional parameters to evaluate thick, coarse-grained clastic reservoirs with significant heterogeneity.

## 4. Results

*4.1. Petrological Characteristics*

The clastic rock reservoir in the study area is mainly composed of coarse lithologies, such as conglomerate, sandy conglomerate, sand-bearing conglomerate, and medium sandstone, with poor gravel sorting and disorderly arrangement (Figure 2a). Gravel is generally contained in the core, with a general gravel diameter of 5–15 mm and a maximum gravel diameter of 30 mm × 50 mm (Figure 2a), sand body thickness of 50–160 m, sand–land ratio of 90%, and single sand body thickness. The reservoir rock type is feldspathic lithic sandstone with a high overall content of 58.4%, mainly medium acid lithic, accompanied by a small amount of sedimentary rock and metamorphic lithic (Figure 2b–e). The sorting difference of the reservoir is shown in Figure 2b, the grinding form is a subcircular to secondary angular shape (Figure 2c), and point-line contact and argillaceous cement are shown in Figure 2d and e, respectively. The structural and compositional maturity of the reservoir is low, primarily because the study area is close to the source area and the transport distance is short. In general, the main characteristics of the reservoir are near provenance deposition, large deposition thickness, complex lithology composition, and strong reservoir heterogeneity.

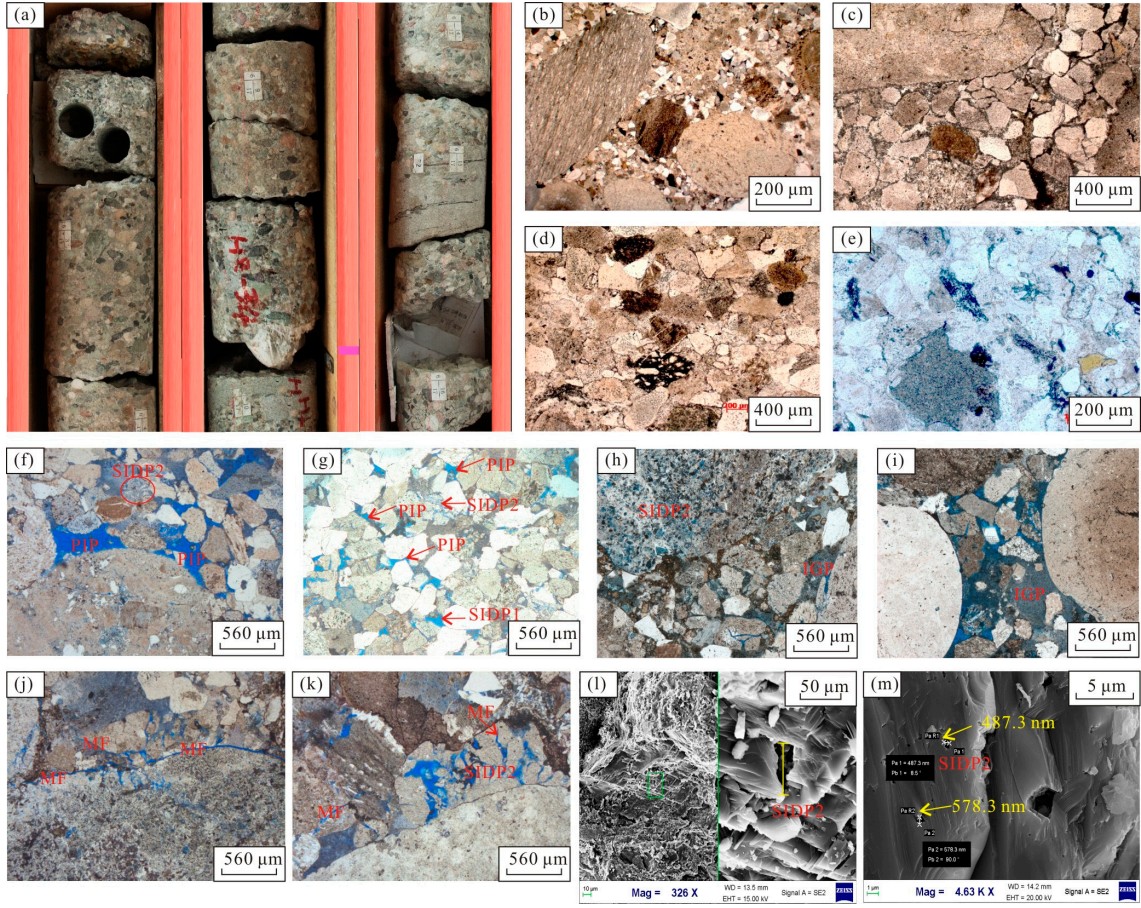

**Figure 2.** Reservoir core and microscopic characteristics of the first member of the Jurassic Badaowan Formation in the Maxi Slope. (**a**) Aihu 501, part of the core between 2831.11 m and 2839.02 m, the core is generally gravel. (**b**) Aihu 501, 2835.32 m, glutenite, poor sorting, particle support, thin section. (**c**) Aihu 501, 2841.81 m, sandy conglomerate, poor sorting, subround to subprism, particle support, thin section. (**d**) Aihu 501, 2840.47 m, medium sandstone, poor sorting, thin section. (**e**) Aihu 501, 2822.84 m, fine sandstone, line contact, casting thin section. (**f**) Ma 625, 2625.16 m, remaining intergranular pores, point-line contacts, casting thin section. (**g**) Ma 606, 2707.48 m, remaining intergranular pores, clear pore edge, casting thin section; (**h**) Aihu 501, 2823.96 m, developed a large number of intragranular dissolved pores and some microcracks, casting thin sections. (**i**) Aihu 501, 2836.16 m, intergranular pore, casting thin section. (**j**) Ma 625, 2836.75 m, structural fracture, casting thin section. (**k**) Ma 625, 2641.4 m, along the direction of the dissolution of microfracture feldspar particles to form secondary pores, casting thin sections. (**l**) Aihu 501, 2823.96 m, feldspar dissolved and formed dissolved pores, enlarged, 326×, SEM. (**m**) Aihu 501, 2847.92 m, feldspar dissolution and formation of intragranular dissolved pores, 4630×, SEM. Note: PIP: primary residual intergranular pores. SIDP1: secondary interparticle dissolution pore. SIDP2: secondary intragranular dissolved pore. IGP: intergranular pore. MF: microfracture.

## 4.2. Physical Characteristics

According to the physical property analysis statistics of clastic rock samples from Aihu 501, the distribution range of reservoir porosity is 3.1–16.3%, with an average value of 9.02%, and the pore concentration distribution range is 6–10% (Figure 3a). The permeability distribution range is $0.014 \times 10^{-3}$–$33.8 \times 10^{-3}$ $\mu m^2$, with an average value of $3.11 \times 10^{-3}$ um$^2$ (Figure 3b). The permeability has a bimodal characteristic, and the permeability concentration distribution range is $0.32 \times 10^{-3}$–$5 \times 10^{-3}$ um$^2$, which indicates that it is a low porosity and low permeability reservoir (Figure 3). The poor correlation between porosity and permeability in the study area indicates that the reservoir pore structure is poor, its

heterogeneity is strong, and the longitudinal physical property changes significantly, which affects the longitudinal differentiation of oil and water.

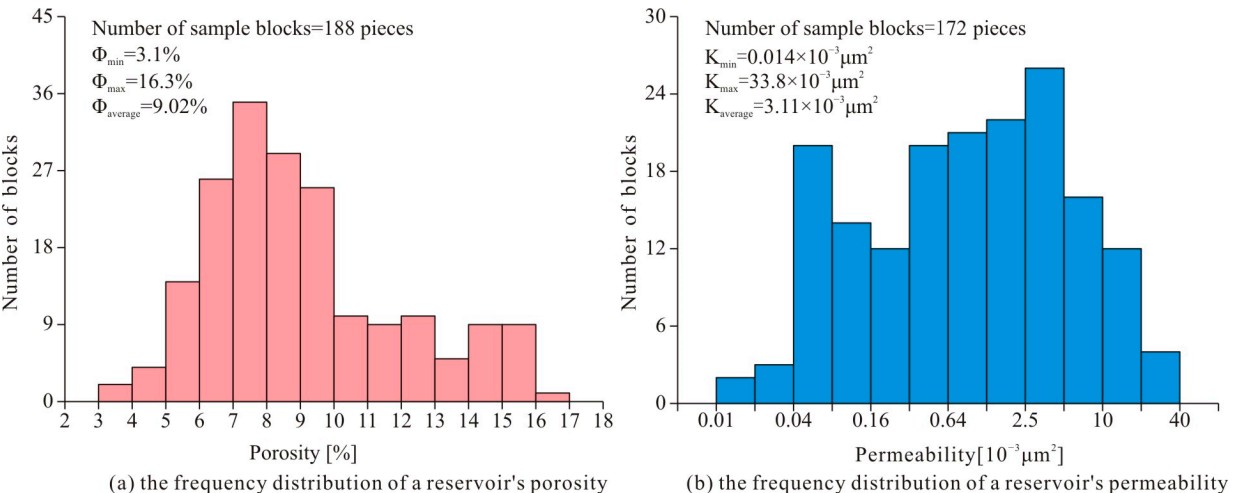

**Figure 3.** Histogram showing the frequency distribution of a reservoir's physical properties in the Badaowan Formation, western slope of the Mahu Depression.

### 4.3. Pore Structure Characteristics

The first member of the Badaowan Formation in the Maxi Slope develops into three main types: primary pores, secondary pores, and microfractures. The primary pores are mainly residual and primary intergranular pores. They generally appear as regular triangles or polygons with clear pore edges (Figure 2f,g). The secondary pores mainly develop intergranular and intragranular dissolved pores and intergranular pores, and the edges of the secondary pores are fuzzy and often irregular (Figure 2h). The high content of intra-granular-dissolved pores are primarily reservoirs with a high content of feldspar and easy solution rock chips, whereas reservoirs, which have a relatively high content of intergranular pores, are mainly composed of kaolinite intergranular pores and micropores in an argillaceous hybrid base (Figure 2i). Microfractures in the study area are mainly structural fractures caused by a tectonic rupture (Figure 2j), compressive fractures caused by the compaction of quartz and other rigid particles in the diagenetic process and dissolution fractures of feldspar particles (Figure 2k), etc. The fractures greatly improve the physical properties of reservoirs. Under scanning electron microscopy, the main clay minerals in the rocks are page kaolinite and vermicular kaolinite, followed by curved lamellar illite. The autogenetic minerals are primarily quartz and carbonate and zeolite minerals, which fill the space between grains, and feldspar with dissolution properties (Figure 2l,m), which can easily form secondary pores. The overall pore development and connectivity of the sample are poor. The mercury injection experiment conducted on the Badaowan Formation in Aihu 501 found that the reservoir had medium displacement pressure, high median pressure, small median radius, and medium average capillary radius. According to the shape of the capillary pressure curve (Figure 4), the pore structure of the reservoir could be divided into two categories: I. good sorting and fine skew degree and II. good sorting and coarse skew degree. The overall performance was medium–fine skew, the sorting was general, and the flow porosity was mainly medium–fine pore throat. To sum up, the pore structure of the reservoir in the study area is poor, and the reservoir heterogeneity is significant.

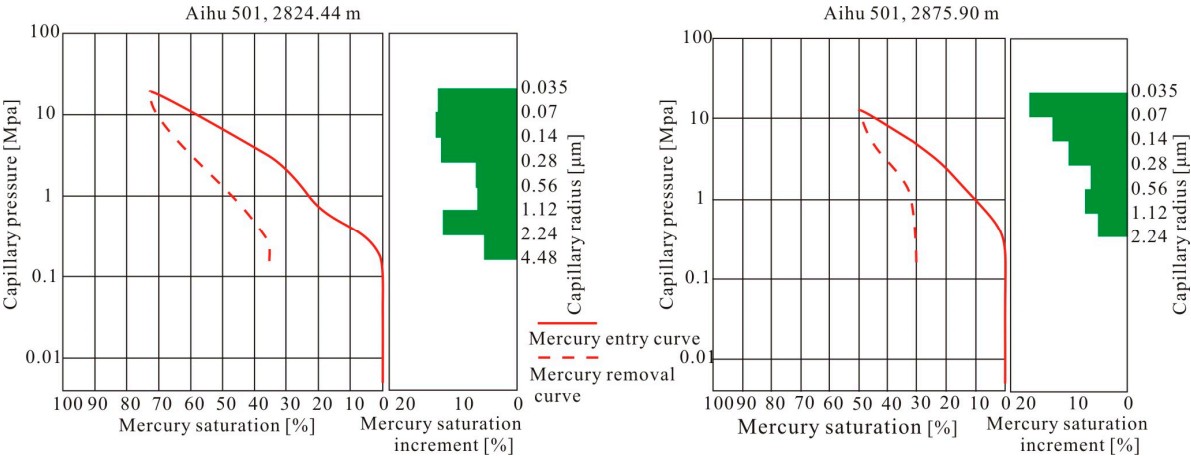

**Figure 4.** Capillary pressure curve of the Badaowan Formation reservoir in Aihu 501.

### 4.4. Reservoir Interpretation Parameter Model

According to the analysis and laboratory data relating to the core well, it was found that there was a good correlation between porosity, density, and acoustic wave in the study area. Models of density and effective porosity (Figure 5a) and acoustic wave and effective porosity (Figure 5b) were established. To improve accuracy, multiple regression processing can effectively avoid excessive values and make the values stable within a certain range. The interpretive model of porosity, density, and soundwave was, therefore, derived using multiple regression (2). The permeability interpretation model (3) was determined using the analysis and laboratory data relating to the core well. The interpretation model was modified by strict use of the core calibration logging in the interpretation process (Figure 6). Porosity and permeability showed an exponential correlation trend. As porosity increased, permeability increased exponentially. The greater the effective porosity, the greater the permeability.

$$Por = 59.97 - 26.67 \times Den + 0.69 \times Cnl \tag{2}$$

$$Per = 0.0008e^{0.616 \times Por} \tag{3}$$

*Por*: porosity (%), *Per*: permeability ($10^{-3}$ um$^2$), *Den*: density (g/cm$^3$), *Cnl*: neutron density logging (%) and *e*: Euler's number.

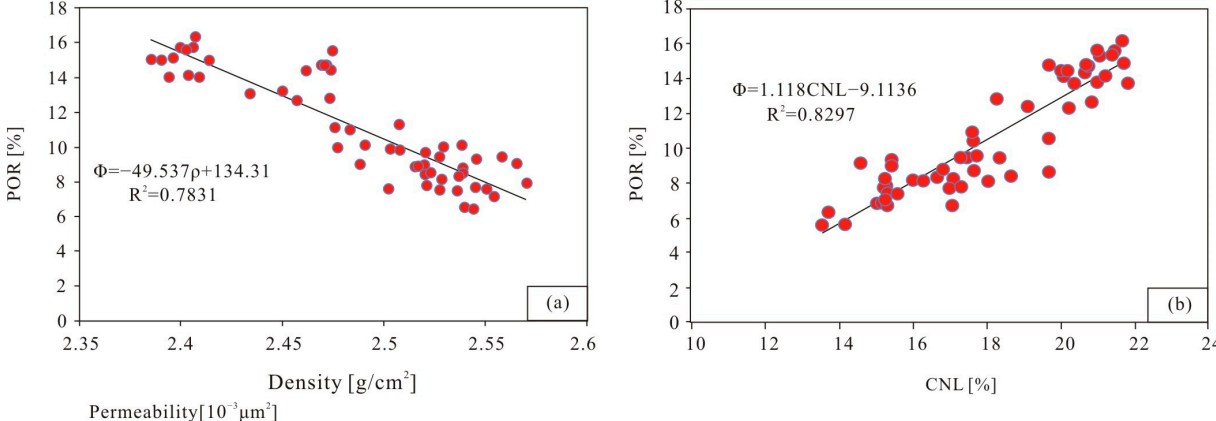

**Figure 5.** Intersection diagram of density, CNL, and effective porosity: (**a**) Intersection diagram of density and effective porosity in the Maxi area and (**b**) intersection diagram of CNL and effective porosity in the Maxi area. R$^2$: determination coefficient.

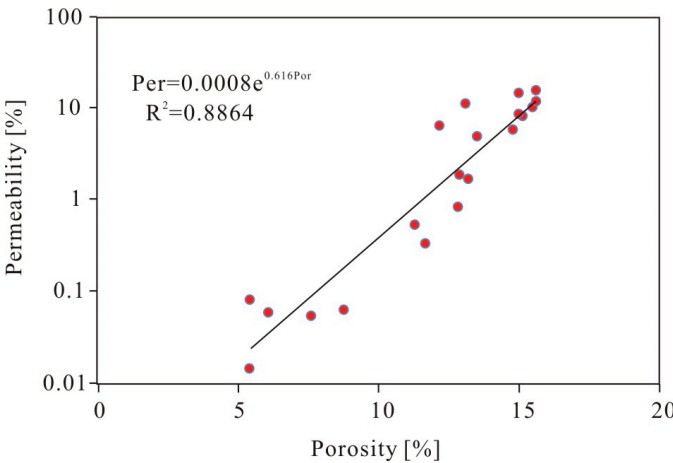

**Figure 6.** Intersection diagram of porosity and permeability in the Maxi Area.

*4.5. Reservoir Heterogeneity Evaluation*

Using the logging interpretation model, the corresponding porosity of each point was calculated, and then the permeability value was obtained. The fluctuation coefficient formula (1) was substituted to calculate the fluctuation coefficient of each well destination layer (Table 1). In the south of the study area, the fluctuation coefficient was large and the heterogeneity strong, and the heterogeneity gradually became weaker from south to north, showing a fan-like distribution. Combined with the oil and production test data, it was found that the larger the fluctuation coefficient in the study area, the stronger the heterogeneity, and the larger the production (Figure 7), which is contrary to the conventional understanding. In this case, based on previous research results, the author believes that because of the thick reservoir and coarse particle size the oil and gas that is formed can escape extremely easily. If an area with strong heterogeneity is encountered, the oil and gas escape can be effectively prevented, so as to accumulate and form a reservoir.

**Table 1.** The fluctuation coefficient calculation data of the Jurassic Badaowan Formation in the Maxi area.

| Well | Starting Depth [m] | Stop Depth [m] | Coefficient of Fluctuation |
|---|---|---|---|
| Aican 1 | 2720.4 | 2727.5 | 0.47 |
| Ma 601 | 2800 | 2812.1 | 0.39 |
| Ma 603 | 2748.2 | 2769.7 | 0.57 |
| Ma 30 | 2764.3 | 2771.9 | 0.18 |
| Aihu 502 | 2783.7 | 2796.7 | 0.27 |
| Aihu 501 | 2824.3 | 2829.4 | 0.46 |
| Aihu 6 | 2817.2 | 2821.6 | 0.25 |
| Aihu 1 | 2768.1 | 2790.1 | 0.44 |
| Ma 611 | 2807.9 | 2821.2 | 0.36 |
| Aihu 5 | 2835.9 | 2847.1 | 0.2 |

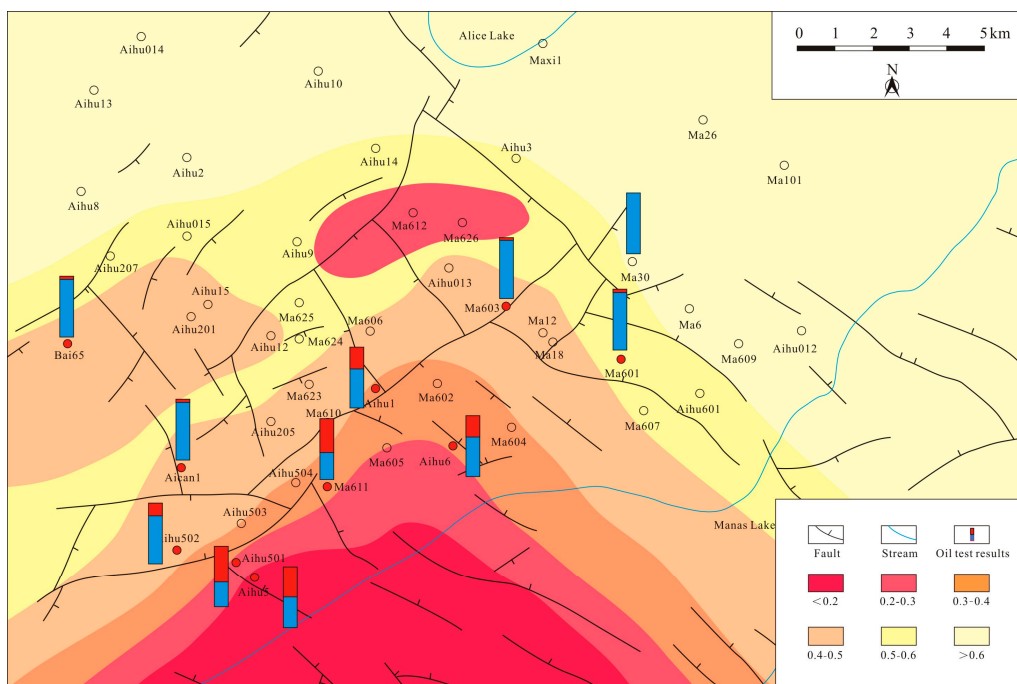

**Figure 7.** Plane distribution of the Jurassic *Badaowan* Formation reservoir heterogeneity in the Maxi area.

## 5. Discussion

### 5.1. Causes of Strong Heterogeneity Reservoir Control

By combining reservoir characteristics and the hydrocarbon accumulation process [30–33], a hydrocarbon accumulation model of "late structural tilting adjustment charging and strong reservoir heterogeneity control" in low-saturation reservoirs of the middle–shallow Jurassic Badaowan Formation in the Maxi Slope was constructed (Figure 8). This demonstrates: (1) a strong hydrodynamic background, frequent erosion and overlaying of multi-stage river channels, irregular distribution of shale interlayers, poor continuity, good reservoir physical properties, and no oil and gas charging; (2) in the early charging period (late J and early K), the Badaowan Formation had a shallow burial depth, good physical properties, the overall reservoir was higher than the average, and the high parts of the structure were pooled; (3) in the reservoir densification stage (K–E), with the increase in burial depth, compaction and cementation led to the escape of some oil and gas, the reservoir became dense and was destroyed, and the oil–water interface began to differ; (4) in the late adjusted charging (E–N), the structure tilted and the Baikouquan reservoir adjusted charging. Under the conditions of low porosity and low permeability, oil and gas migration and accumulation were controlled by strong heterogeneity and were widely distributed and disjointed, forming a low-saturation reservoir [34–37]. The discovery of the low-saturation oil and gas reservoir expanded the area of oil and gas exploration in the Mahu Depression, triggered similar oil and gas exploration in the Junggar Basin, and strengthened the exploration case for locating low-saturation oil and gas reservoirs in thick sand conglomerate low-permeability reservoirs.

The heterogeneity of thick, coarse-grained clastic rock reservoirs is significant because of the complex and varied rock composition, variability in rock particle size, low maturity, poor sorting, significant difference in vertical physical properties, and the complex sedimentary structure. Conventional evaluation parameters are not applicable, and their defects were discussed in the previous paper. Because of the geological characteristics of thick, coarse-grained clastic rock reservoirs, use of the fluctuation coefficient is proposed to evaluate their heterogeneity.

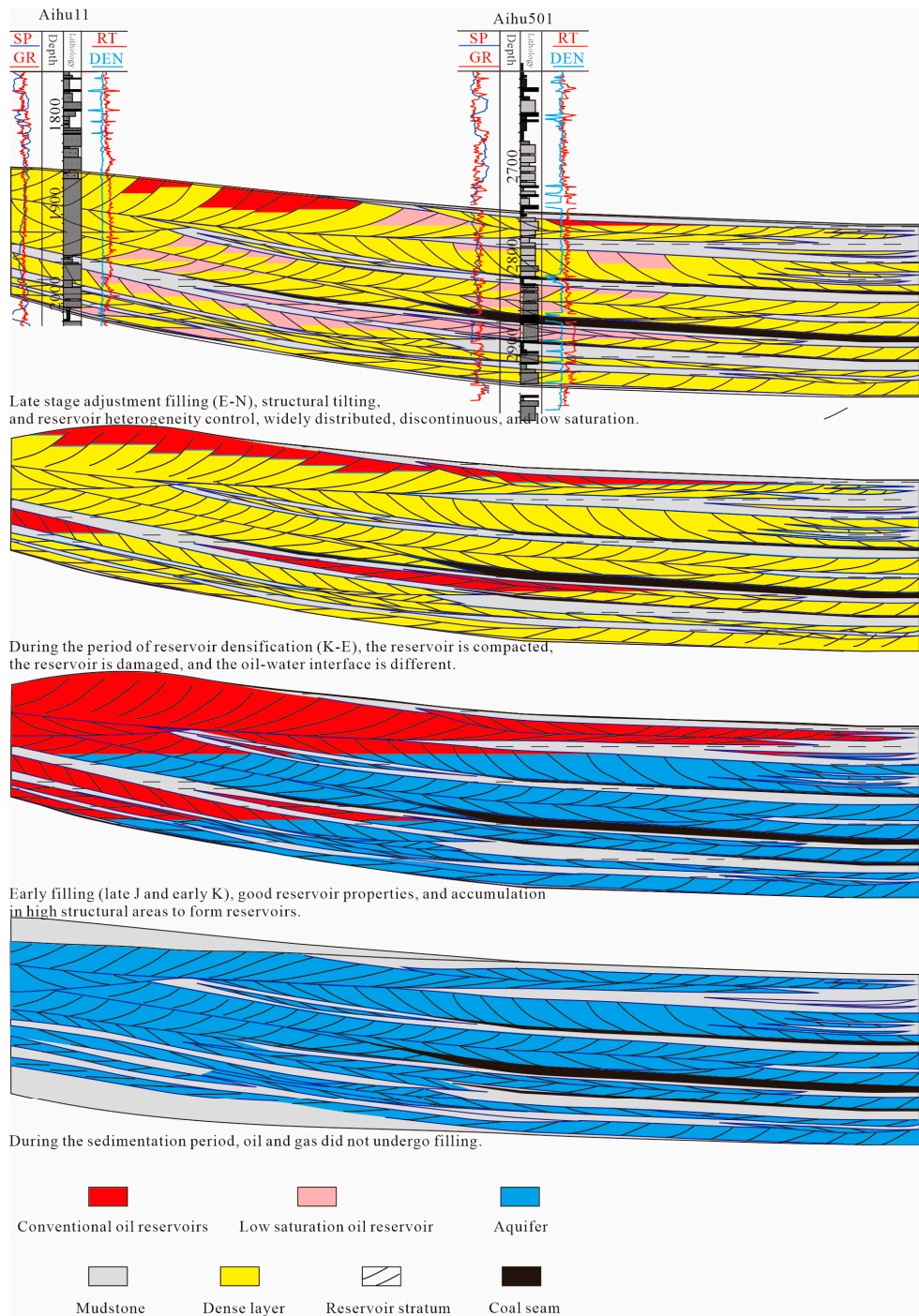

**Figure 8.** Oil and gas *accumulation* model of the first member of the Badaowan Formation in the Maxi region.

## 5.2. Geological Significance of the Fluctuation Coefficient

Based on the study of cumulative distribution characteristics of the physical property parameters of reservoirs, the wave coefficient represents a new parameter for heterogeneity characterization. Compared with conventional heterogeneity evaluation parameters, the wave coefficient assesses reservoir heterogeneity more comprehensively and accurately [38,39]. It has the characteristics of bounded calculation data, a clear characterization target and unified characterization angle, etc., and avoids the drawbacks of subjective weighting and the lengthy and complicated calculation process caused by data classification. The introduction of the fluctuation coefficient to some extent alleviates the lack of

quantitative characterization parameters in the field of thick-layer, coarse-grained clastic rock reservoirs. This new parameter can not only be used for objective evaluation of interlayer heterogeneity, but also can be applied to the evaluation of plane heterogeneity. The fluctuation coefficient is proposed for thick, coarse-grained clastic rock reservoirs with strong heterogeneity. This parameter is suitable for reservoir evaluation in the Badaowan Formation in the Junggar Basin; its calculation and operability are simple, and it can be widely used.

## 6. Conclusions

(1) Taking the characteristics of thick reservoir heterogeneity into consideration, reservoir evaluation parameters were fully explored, and the fluctuation coefficient is proposed for the evaluation of reservoir heterogeneity. This essentially involves comparing the average permeability of two adjacent layer sites with the average permeability. The fluctuation coefficient reflects the fluctuations in permeability and has the advantages of a clear characterization target, bounded calculation data, and the same characterization angle, so as to realize the quantitative characterization of the degree of macroscopic heterogeneity of the reservoir under a unified standard. The larger the fluctuation coefficient, the stronger the heterogeneity, and vice versa.

(2) The pore structure of thick, coarse-grained clastic rock reservoirs is poor, heterogeneity is strong, and the overall performance of the reservoir is "good porosity and low permeability". This parameter is used to evaluate reservoir heterogeneity. The fluctuation coefficient of most wells in the Maxi area is about 0.3, but some wells can reach 0.2 and a few are at about 0.6, demonstrating that the larger the fluctuation coefficient, the stronger the heterogeneity and the better the oil content. Unlike conventional reservoirs, these are heterogeneous and there is strong reservoir control.

(3) The fluctuation coefficient is a new parameter for heterogeneity characterization, which is derived from the physical property characteristics of the reservoir. It not only improves the current parameter system for reservoir heterogeneity characterization, but also resolves the problem of the lack of quantitative characterization parameters and methods for in-layer and in-plane heterogeneity. The fluctuation coefficient can effectively evaluate the strength of reservoir heterogeneity and can provide a reference for reservoir research in areas with similar geological conditions.

**Author Contributions:** Conceptualization, B.Z. and X.Z.; methodology, B.Z.; validation, X.J. and X.R.; formal analysis, X.J.; investigation, X.W.; resources, W.Z.; data curation, X.J.; writing—original draft preparation, B.Z.; writing—review and editing, X.Z.; visualization, X.J.; supervision, X.Z.; project administration, X.Z.; funding acquisition, X.Z. All authors have read and agreed to the published version of the manuscript.

**Funding:** This research was funded by the Karamay innovative environment construction program -innovative talent project, grant number 20232023hjcxrc0049 and the Young Natural Science Foundation of Xinjiang Province, China, grant number 2021D01F39.

**Data Availability Statement:** Not applicable.

**Conflicts of Interest:** The authors declare no conflict of interest.

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
