# Peer review of "Establishment and Application of a New Parameter Model for Quantitative Characterization of the Heterogeneity of Thick, Coarse-Grained Clastic Reservoirs: A Case Study of the Badaowan Formation in the Western Slope of the Mahu Depression, Junggar Basin, China"

_processes, doi:10.3390/pr11082423_

Round 1
Reviewer 1 Report
This paper includes scientific reserch of the reservoir heterogeneity characterization including parameters of perme- 145 ability variation coefficient, permeability rush coefficient and permeability contrast, therefore I suggest publication.
Author Response
Dear Reviewers:
Thank you for your comments concerning our manuscript entitled “Establishment and application of a new parameter model for quantitative characterization of the heterogeneity of thick, coarse-grained clastic reservoirs: a case study of the Badaowan formation in the western slope of the Mahu Depression, Junggar Basin, China”. Those comments are all very valuable and very helpful for revising and improving our manuscript. These comments also have guiding significance to our later scientific researches. We have studied these comments carefully and revised and corrected our MS according to each comment, which we hope meet with approval. And revised portion are marked in yellow highlight in the paper. The main corrections in the paper and the answers to the reviewer’s comments are as following:
Answers to reviewer 1
1.This paper includes scientific research of the reservoir heterogeneity characterization including parameters of perme- 145 ability variation coefficient, permeability rush coefficient and permeability contrast, therefore I suggest publication.
Answer: Thank you very much for your endorsement of our MS.
Special thanks to you for your valuable comments.
Reviewer 2 Report
The abstract discusses the investigation of complex rock composition and heterogeneity in thick-layer coarse-grained clastic reservoirs in the Baodaowan Formation of the Junggar Basin, China. However, it appears that most of the work has been previously published, which raises concerns about the originality and contribution of this study. Additionally, the abstract mentions that the reservoir is extremely heterogeneous, but the explanation of this heterogeneity and its significance is not clearly discussed. The abstract lacks in-depth details about the specific results and findings, which diminishes the impact of the study.
Another concern is the quality of the figures. High-quality figures are crucial for the proper understanding and interpretation of research results. If the figures in the manuscript are of poor quality, it could significantly affect the overall presentation and value of the research.
To strengthen the manuscript as a whole, the authors should provide more details on the unique aspects and contributions of this study compared to previously published work. Additionally, improving the quality of the figures is essential to enhance the visual clarity and presentation of the research results.
Overall, while the topic is relevant, the abstract's lack of detailed information and originality, combined with concerns about figure quality, raise questions about the overall quality and novelty of the study. The authors should carefully address these issues to improve the manuscript's impact and suitability for publication.
A rejection decision is recommended.
Extensive editing of English language required
Reviewer 3 Report
This article focused on coarse clastic rock reservoir, and a new quantitative characterization parameter of heterogeneity based on the average distribution difference characteristics of reservoir physical parameters is proposed, and the fluctuation coefficient of thick coarse-grained clastic rock reservoir is calculated, the reservoir heterogeneity is evaluated. There are a few questions to be addressed,
(1) First of all, English of this manuscript should be improved. Introduction was not well organized and should be rewritten to clearly illustrate the research results or data about the topic. In addition, what does domestic means, refer to China? The introduction should include all the research results, not limited to the region.
(2) The new quantitative characterization parameter of heterogeneity based on the average distribution difference characteristics of reservoir physical parameters is proposed in the manuscript. This parameter is the accumulation of permeabilizes divided by its average value, and it cannot exclude the possible conditions that exists in Fig.1 Please explain the specific conditions it fits and compared it with traditional ones to show advantage of it. Experimental results can be added to further confirm your conclusion.
(3) The manuscript should be more focused on the topic. Current version only includes one part related to quantitative characterization parameter. Reservoir interpretation parameter model developments compose the large part of content. Then either the manuscript structure should be modified, or the topic be renamed.
English of this manuscript should be improved.
Reviewer 4 Report
Manuscript
Title: „Quantitative Characterization of Heterogeneity in Thick Coarse-grained Clastic Reservoirs: A Case Study of the Badaowan Formation in the Western Slope of Mahu Depression, Junggar Basin, China”
Authors: Boyu Zhou, Xiaodong Zhao, Xuebing Ji, Xinyu Wu, Wenping Zhao and Xi Rong.
Dear Authors
I revised the manuscript: "Quantitative Characterization of Heterogeneity in Thick Coarse-grained Clastic Reservoirs: A Case Study of the Badaowan Formation in the Western Slope of Mahu Depression, Junggar Basin, China” submitted to the „Processes” Journal. The paper is very interesting. However, I have some concerns, which need to be addressed.
Line 2-5. Article topic
The theme of the article is concise and accurately reflects the content of the article.
The structure of the article, divided into chapters and subchapters, is clear and logical. The theme of the article is specific to the case study.
Abstract
Line 14-43. The content of the abstract should indicate the measurable effect of the research in terms of giving a numerical value and indicating the most important conclusions generalised even within the implementation of the case study. Please take this into account.
The abstract is a self-contained part of the article which requires repetition of any explanatory notes.
The authors suggest the goal and scope of the work, but the main goal of the work is split between individual specific tasks, which are shown implicitly as sub-results. The descriptions of the results are very abbreviated.
It is difficult to identify the leading result and the leading conclusion of the research. Please take this into account.
Keywords. Line 44-45. The authors used the full spectrum of matching wording. Keywords contain word clusters and are too literal, but represents the spectrum of information well. The order of keywords should follow the concept of „from generalities to specifics”. Keywords may also reflect the order in which the research issues are addressed. Please consider changing the order of keywords.
Please try to arrange your keywords according to a coherent concept and dispense with near-meaningful expressions and duplicate messages.
1. Introduction
The state of the knowledge presented is relevant to the goal and scope of the research. The presentation of the state of knowledge in the chapter is quite general and its form is quite brief. Please consider elaborating the content provided in more detail in the context of an attractive and internationally recognised research topic.
The included information indicates research results related to a case study analysis (case study), with limited possibilities of generalisation. This is a conclusion and should conclude the analysis of the research results instead making a form of hypothesis. If the research result allows for its comprehensive use then please articulate this.
2. Limitations of conventional heterogeneous characterization parameters
Dividing the state of the knowledge analysis into two independent chapters is not necessary. Please take this into account.
Line 144. Figure 1.
Permeability/10-3 μm2 . - The "slash"/sign is related to division and confuses the reader. If the designation relates to a unit of measurement then brackets should be used, for example: [10-3 μm2].
Depth/m – The "slash"/sign is related to division and confuses the reader. If the designation relates to a unit of measurement then brackets should be used, for example: Depth [m].
3. Materials and Methods
Line 147 – 193. The information is presented in a structured manner and is adequate to the scientific issue addressed.
Line 154 – 155. „…Wells Aihu 501, Ma 625, 154 Ma 606 and Bai65….” Please explain the introduced abbreviations of terms, possibly as soon as they are introduced in the content of the article. Even popular acronym names need to be explained in scientific articles. A single explanation is sufficient in the execution of the explanation.
Line 159. „…Well Aihu 501….” Please explain the introduced abbreviations of terms, possibly as soon as they are introduced in the content of the article. Even popular acronym names need to be explained in scientific articles. A single explanation is sufficient in the execution of the explanation.
Line 160, 224, 225. „….SEM….” Please explain the introduced abbreviations of terms, possibly as soon as they are introduced in the content of the article. Even popular acronym names need to be explained in scientific articles. A single explanation is sufficient in the execution of the explanation.
Line 163. „…HD-505…” Please explain the introduced abbreviations of terms, possibly as soon as they are introduced in the content of the article. Even popular acronym names need to be explained in scientific articles. A single explanation is sufficient in the execution of the explanation.
Line 165. „…30Mpa….” Lack of space between parameter value and unit of measurement. Please correct this. For example 30_MPa.
Line 165. „…0.025um….” Lack of space between parameter value and unit of measurement. Please correct this. For example 0.025_μm.
Line 186- 188. Mathematical equation 1. Lack of units of measurement for model components in mathematical notation. Please complete this.
Parameter „vb” - Lack of description and reference to the unit of measurement.
4. Results
Line 199. „….5-15mm….” Lack of space between parameter value and unit of measurement. Please correct this. For example 15_m. The unit of measure should be present with both range values. Both range values should have the unit of measurement „m”.
Line 200. „…30 mm × 50mm….” Lack of space between parameter value and unit of measurement. Please correct this. For example 50_m.
Line 200. „…50-160m….” Lack of space between parameter value and unit of measurement. Please correct this. For example 160_m. The unit of measure should be present with both range values. Both range values should have the unit of measurement „m”.
Line 212. „…Aihu 501 well…” Change order of notation elements or new parameter. Please unify the notation of the individual parameter names. Any change in notation is only justified by the introduction of a new parameter.
Line 212, 213, 215, 216, 217, 218, 219, 220, 221, 223, 224, . „….2831.11m….” Lack of space between parameter value and unit of measurement. Please correct this. For example 2831.11_m.
Line 221. „….Ma625 well….” Lack of spaces in parameter notation. Please unify parameter notation due to the need for readers to understand the text correctly.
Line 225. „….PIP…” primary residual intergranular pores? The acronym does not exactly match the textual explanation. Please explain the difference as the acronym PIP has many meanings different from the one indicated by the authors.
Line 230, 231, 232, 234. „….~…..” A hyphen with a special character is understood as " from - to" with and without the use of a space. Please unify the range notation.
Line 239, 240, 241. Figure 3. Lack of references to the designations of graphic forms (a) and (b) in the text of the dedicated subsection and in the description of Figure 3.
Line 246. „…Figure 2f-g….” The description is not unambiguous. Please try entering a space or repeating the numerical designation for range values. For example: 2f-2g
Line 260 „….Figure 2l-m…..” The description is not unambiguous. Please try entering a space or repeating the numerical designation for range values. For example: 2l-2m
Line 270. Figure 4. Text description in graphic area is unreadable. Font too small.
Line 262, 271. „…Well AiHu 501. ….” The written letter notation made with upper and lower case characters, separated by a space, for example: Aihu, Ai Hu and AiHu are substituted intentionally or accidentally? Please homogenise the notation.
Line 271. „…Well Ai Hu 501…..” The notations in Figure 4 and in thedescription to Figure 4 are different. Please explain the differences.
Line 284 – 288. Figure 5. Mathematical formula (model).
„….Por….” Lack of description of the acronym used and the unit of measurement. For example: porosity
„….Per…..” Lack of description of the acronym used and the unit of measurement. For example: permeability
„….den…..” Please explain the introduced abbreviations of terms, possibly as soon as they are introduced in the content of the article. Even popular acronym names need to be explained in scientific articles. A single explanation is sufficient in the execution of the explanation. For example: Density
„….CNL…” Cnl - Please explain the introduced abbreviations of terms, possibly as soon as they are introduced in the content of the article. Even popular acronym names need to be explained in scientific articles. A single explanation is sufficient in the execution of the explanation.
„….e…..” Please explain the introduced abbreviations of terms, possibly as soon as they are introduced in the content of the article. Even popular acronym names need to be explained in scientific articles. A single explanation is sufficient in the execution of the explanation. For example: Euler's number
„…..R2…..” Please explain the introduced abbreviations of terms, possibly as soon as they are introduced in the content of the article. Even popular acronym names need to be explained in scientific articles. A single explanation is sufficient in the execution of the explanation. For example: determination coefficient
Line 304. Table 1. „…Starting depth….Stop depth….” Lack of unit of measurement for the values indicated. Please complete this.
5. Discussion
Line 336. „…[38-39]….” A range notation is not necessary in the situation of directly consecutive literature items. The notation [38,39] is therefore sufficient. Please consider this.
A discussion of the results of the research is present although the authors interpret the results of the cited articles according to the thesis indicated. Please consider presenting a wider context for alternative solutions to the research question and, if none exist, indicate the uniqueness of the new and proposed method.
6. Conclusions
The authors build explanations for the research results obtained and indicate the methodological aspect of increasing scientific knowledge within the discipline. Conclusions are presented briefly but reflect well on the research findings. Please consider presenting the practical implications of the results obtained going beyond the case study.
References
Line 376 - 469. The bibliographic description should be supplemented, if possible, by the publication reference, for example: https://doi.org...... Please take this into account.
Please avoid citations of academic textbooks and non-peer-reviewed professional publications.
Reviewer 5 Report
Review of the paper by Zhou et al, Quantitative Characterization of Heterogeneity in Thick Coarse.Grained clastic reservoir.
The paper looks at a new parameter to evaluate heterogeneity in coarse-grained clastic reservoir.
Comments:
The abstract should be rewritten. It is not clear, not appealing and contains also too long phrases.
Introduction: it is not clear if this new parameter can be applied to characterize this specific Badahowan Formation or all coarse grained clastic reservoir.
Which is the interest for readers? Please, rewrite also the introduction stressing the importance of this new parameter (if it is so) and application in similar cases.
Overall the paper is not interesting to the reader and should be improved. It reports a study case useful to oil companies working on this formation but not interesting for scientists all over the world.
My advice is to accept after major revision.

English is poor and should be revised by mother language writer.
Round 2
Reviewer 2 Report
I have no additional remarks to make. The manuscript is prepared for acceptance.
Reviewer 3 Report
The author has made revisions based on reviewer`s suggestions.
English has been improved.
Reviewer 5 Report
Dear Authors,
I had a careful look at the revised version of your manuscript and I see that you have modified the text according to suggestions.
The paper is now more interesting, the aims and conclusions have been better addressed and achieved, and the English language has been improved.
Consequently, I think the manuscript, entitled "Quantitative Characterization of Heterogeneity in Thick Coarse-grained Clastic Reservoirs: A Case Study of the Badaowan Formation in the Western Slope of Mahu Depression, Junggar Basin, China” by Boyu Zhou, et alii, now fully deserves to be published.